# Relationship of Diet to Gut Microbiota and Inflammatory Biomarkers in People with HIV

**DOI:** 10.3390/nu14061221

**Published:** 2022-03-14

**Authors:** Mónica Manzano, Alba Talavera-Rodríguez, Elena Moreno, Nadia Madrid, María José Gosalbes, Raquel Ron, Fernando Dronda, José A. Pérez-Molina, Val F. Lanza, Jorge Díaz, Santiago Moreno, Beatriz Navia, Sergio Serrano-Villar

**Affiliations:** 1Department of Nutrition and Food Science, Faculty of Pharmacy, Universidad Complutense de Madrid, 28040 Madrid, Spain; 2Department of Infectious Diseases, Hospital Universitario Ramón y Cajal and IRYCIS, 28034 Madrid, Spain; albatalavera92@gmail.com (A.T.-R.); emolmo@salud.madrid.org (E.M.); nadiapatricia.madrid@salud.madrid.org (N.M.); raquel.rongonzalez@gmail.com (R.R.); fdronda@hotmail.com (F.D.); jperezm@salud.madrid.org (J.A.P.-M.); jolohinodam@gmail.com (J.D.); smoreno.hrc@salud.madrid.org (S.M.); 3CIBERINFEC, Instituto de Salud Carlos III, 28029 Madrid, Spain; 4FISABIO-Salud Pública, 46020 Valencia, Spain; maria.jose.gosalbes@uv.es; 5CIBERESP, Instituto de Salud Carlos III, 28029, Madrid, Spain; 6Bioinformatics Unit, Hospital Universitario Ramón y Cajal and IRYCIS, 28034 Madrid, Spain; valfernandez.vf@gmail.com; 7Research Group VALORNUT-UCM (920030), Universidad Complutense de Madrid, 28040 Madrid, Spain

**Keywords:** HIV, diet, microbiota, *Lachnospira*, Erysipelotrichaceae, inflammatory biomarkers, D-dimer, TNF

## Abstract

While changes in microbiome composition have been associated with HIV, the effect of diet and its potential impact on inflammation remains unclear. Methods: Twenty-seven people living with HIV (PWH) on antiretroviral therapy (ART) were studied. A comprehensive dietary analysis was performed and two types of dietary patterns were determined. We explored the associations of each dietary pattern with gut microbiota and plasma inflammatory biomarkers. Results: We appreciated two dietary patterns, Mediterranean-like (MEL) and one Western-like (WEL). Compared to participants with the WEL pattern, participants with MEL pattern showed higher abundance of *Lachnospira* (*p*-value = 0.02) and lower levels of the inflammatory biomarkers D-dimer (*p*-value = 0.050) and soluble TNF-alpha receptor 2 (sTNFR2) (*p*-value = 0.049). Men who have sex with men (MSM) with MEL pattern had lower abundance of Erysipelotrichaceae (*p*-value < 0.001) and lower levels of D-dimer (*p*-value = 0.026) than MSM with WEL pattern. Conclusion: MEL pattern favours *Lachnospira* abundance, and protects against Erysipelotrichaceae abundance and higher levels of the inflammatory biomarkers D-dimer and sTNFR2, precursors of inflammatory processes in HIV-infected patients. Our study contributes to understanding the determinants of a healthier diet and its connections with gut microbiota and inflammation.

## 1. Introduction

Interactions between altered gut mucosa and bacteria during HIV infection appear to contribute to chronic immune dysfunction and correlate with the route of HIV transmission [1,2]. However, further study is needed to understand how nutritional interventions might improve gut dysbiosis, especially during chronic diseases.

Scientific evidence has associated the more Western dietary pattern, high in saturated fatty acids, cholesterol and sugars and low in fibre and micronutrients, with various inflammatory metabolic and chronic gastrointestinal tract conditions, whereas diets rich in fibre, vitamins and antioxidants have beneficial effects on gut homeostasis by increasing microbial diversity and inducing a regulatory environment [3,4,5,6]. Although diet and dietary habits have been extensively studied in healthy people or people with different pathologies, studies referring to diet in HIV-infected patients are very scarce [6,7].

On the other hand, some studies have found differences in the intestinal microbiota of HIV-infected patients by transmission risk group and have suggested that these differences could be attributed to diet [1,8].

We aimed to analyse the dietary patterns and dietary quality of a group of 27 people living with HIV (PWH), and correlate the nutritional parameters with the gut microbiota composition and inflammatory biomarkers.

## 2. Materials and Methods

### 2.1. Study Design and Sample Collection

We conducted a cross-sectional study. Participants were recruited at the HIV unit of the Ramón y Cajal University Hospital in Madrid, Spain, between October 2017 and February 2018. Participant contact took place at the HIV unit of the hospital, where physicians informed participants about the details of the study beforehand.

The protocol was approved by the Ethics Committee of the Hospital Ramón y Cajal (Ref. 030/20) and all participants signed an informed consent form.

Stool and blood samples were collected on the same day that the dietary questionnaires were completed.

Inclusion criteria were confirmed HIV infection, age of 18 years or more, and antiretroviral therapy (ART) with undetectable plasma HIV RNA during at least 48 weeks. Exclusion criteria were failure to provide signed informed consent, acute and intercurrent health problems.

### 2.2. Dietary Data

A three-day dietary record [9], including two working days and one weekend day, was used to collect all food, beverages and food supplements taken by the participant during that time period.

The questionnaire was completed by the participants. To ensure that the questionnaire was accurately completed, participants were provided with written detailed instructions on how all information was to be recorded, including food and dish ingredients (where possible), cooking techniques, brand names of products and quantities, which could be recorded in home measurements.

To minimize errors, all data were reviewed by the research group’s nutritionists to identify unrealistic servings or fluid intake or any errors in recording.

All dietary information was processed with the DIAL software version 3.0.0.12 (Alce Ingeniería, Madrid, Spain), which uses data from the Spanish Food Composition Tables [10]. The observed energy intake, the caloric profile of the macronutrients in the participants’ diets, as well as the intake of vitamins and minerals were obtained through this programme [11].

### 2.3. Gut Microbiota

#### 2.3.1. Purification of Nucleic Acids

Faecal samples from participants were stored in Omnigene Gut kits (DNA Genotek, Kanata, ON, Canada). Faecal samples were divided into aliquots and cryopreserved at −80 °C until use.

#### 2.3.2. Amplification of the 16 S rRNA Gene

Total DNA was extracted from faecal samples on the MagNA Pure LC Instrument robotic workstation (Roche, Basel, Switzerland) using the MagNA Pure LC III DNA isolation kit (Bacteria, Fungi) (Roche). Total DNA was quantified with a Qubit fluorometer (ThermoFisher, Waltham, MA, USA). For each sample, regions V3–V4 of the 16 S rRNA gene were amplified and amplicon libraries were constructed following Illumina instructions (Illumina, San Diego, CA, USA) [12]. Sequencing was performed using the V3 kit (2 × 300 cycles) with MiSeq sequencer (Illumina, San Diego, CA, USA) at the Sequencing and Bioinformatics Service FISABIO, Valencia, Spain. We obtained an average of 62,939 united 16 S rRNA sequences per sample.

#### 2.3.3. Pre-Processing and Quality Control

All sequences used in this analysis passed quality control, where the length and quality of reads were filtered using trimmomatic v0.33. To standardise the number of reads in the diversity analyses, sub-sampling methods (seqkit, sample subcommand) were used, which was performed based on the minimum number of reads per sample.

#### 2.3.4. 16 S RNA Gene Analysis

The 16 S rRNA gene amplicon data were analysed using the Kraken taxonomic sequence classifier (v2.0.7-beta, paired-end option) [13,14], which examines the k-mers within a query sequence and uses the information within those k-mers to query a database. That database maps k-mers to the lowest common ancestor of all genomes known to contain a given k-mer. Taxonomic information on 16 S rDNA sequences was obtained using the Silva ribosomal RNA database (version 132) [15] available on Kraken 2 web [16]. After assigning taxonomic labels to the sequence reads, the Operational Taxonomic Units (OTU) table was extracted using Pavian version.

### 2.4. Inflammatory Biomarkers

A fasting venous blood sample was collected from each participant and plasma levels of 8 inflammatory biomarkers were determined from the cryopreserved plasma by immunoassay in triplicate. Tumour soluble necrosis factor sTNFR2 (DRT200, R & D Systems, Bio-Techne Corporation, Minneapolis, MN, USA), C-reactive protein (CRP) (DCRP00, Quantikine ELISA kit, R & D Systems, Minneapolis, MN, USA), sCD14 (AbClonal, Wuhan, China), sCD163 (AbClonal, Wuhan, China), FABP2/IFABP (Boster Biological Technology, Wuhan, China), D-dimers (Ray Biotech, Norcross, GA, USA), LTA (Abbexa, Cambridge, UK), LBP (Boster Biolo-gical Technology, Wuhan, China).

### 2.5. Statistical Analysis

Results are presented as mean ± standard deviation (SD) or as re-counts and proportions in the case of categorical variables. Differences between the different dietary patterns were considered statistically significant if *p* < 0.05. To study the normality of the different variables within the sample and within the different analysis groups, the Shapiro–Wilk test recommended for *N* < 50 was used. Comparison of data between dietary patterns was performed using the Student’s *t*-test for two independent samples, in the case of normally distributed variables, and the Mann–Whitney U-test, in the case of non-parametric variables. For nominal variables, the chi-square test (χ^2^) was used.

Dietary patterns were obtained by performing a K-means cluster statistical analysis, prefixing the number of clusters to 2, and standardising (z-scores) the intake variables of 14 food groups measured in total grams per day.

Spearman correlations were used to assess the relationship of the different nutritional variables with gut microbiota and inflammatory biomarkers. The statistical programme IBM SPSS Inc. Version 20.0 (Armonk, NY, USA) was used for the analysis of the results and R software (R Core Team, Vienna, Austria) for the correlation analyses between MEDDQI and bacterial counts (libraries corrplot) and for the rest graphics Orange Data Mining Version 3.30 (University of Ljubljana, Ljubljana, Slovenia) and Ms Office Profesional Plus 2016 Version 21.12 (Microsoft, Redmond, WA, USA).

### 2.6. MED-DQI Dietary Quality Index

To assess the dietary quality of the dietary patterns, the value of the MED-DQI dietary quality index was calculated for each pattern, and for each participant, as it is considered one of the most suitable indices for assessing the quality of the Mediterranean diet in adults [17,18,19]. The calculation was made taking into account the following components: % of saturated fatty acids in relation to total energy, cholesterol (mg), grams of meat, mL of olive oil, grams of fish and grams of fruit and vegetables.

Each nutrient or food group was assigned three scores (0, 1 and 2) based on the recommended guidelines.

MED-DQI scores between 1–4 are considered good, between 5–7 medium good, between 8–10 medium poor and 11–14 poor.

## 3. Results

### 3.1. Description of the General Population

The final sample analysed was 27 participants, 18 of whom were men who had sex with men (MSM), 6 ex-injection drug users (IDU) and 3 heterosexuals (HTX) with a mean age of 48.5 years and of Caucasian, Latin American and Sub-Saharan African origin, all participants were on triple ART and only six participants had taken antibiotics in the 6 months prior to the start of the study (the last intake being 14 weeks before the start of the study) and mean BMI was 25.19 (Table 1). Two dietary patterns were identified as a result of the k-means cluster statistical analysis: the Mediterranean-like pattern (MEL) and the Western-like pattern (WEL).

### 3.2. Dietary Patterns and Dietary Quality Assessment (MED-DQI)

After processing the dietary information, daily intake values (total grams per day) were obtained for 14 food groups (Table 2).

Two dietary patterns were identified as a result of the k-means cluster statistical analysis: the Mediterranean-like pattern (MEL) and the Western-like pattern (WEL). Figure 1 shows the resulting heatmap of the z-score variables for the 14 food groups. 

The MEL pattern was characterised by a higher consumption of vegetables, cereals and legumes, and a lower consumption of sugars, white and red meat and potatoes compared to WEL pattern. The values of the final cluster centres are shown in Table A1.

MEL pattern was characterised by a lower intake of saturated fatty acids (SFA) and a high intake of total dietary fibre (33.99 ± 13.45 g/day vs. 22.23 ± 5.84 g/day of WEL pattern; *p*-value = 0.015), and a higher caloric intake from carbohydrates (Table A2). Regarding the contribution of vitamins, antioxidants and fatty acids, a higher intake of folic acid, biotin, total tocopherols, total carotenes, β-carotenes, and a higher oxygen radical absorbance capacity (ORAC) index was observed (Table A3), as well as a lower intake of stearic acid (5.31 ± 1.98 g/d vs. 8.23 ± 3.81 g/d; *p*-value = 0.044) and arachidonic acid (0.16 ± 0.13 g/d vs. 0.31 ± 0.20 g/d; *p*-value *=* 0.015). 

To assess the dietary quality of the standards, the mean value of the MED-DQI quality index was calculated (Table A2), showing differences, with a mean score of 4.5 for the MEL dietary standard, recognising it as a good-medium good diet, compared to a mean score of 7.5 for the WEL dietary standard, considering it as a medium-poor diet.

### 3.3. Factors Associated with the Risk Group

72% of MSM participants followed MEL dietary pattern compared to 28% of MSM participant who followed WEL dietary pattern. The rest of the non-MSM participants (HTX and IDU) were more evenly distributed, with 44% following MEL pattern and 56% following WEL pattern.

### 3.4. Association between Dietary Pattern and Gut Microbiota

After searching for possible associations between dietary pattern and gut microbiota, we found increased 5-fold *Lachnospira* abundance associated with MEL pattern (*p*-value *=* 0.02) (Figure 2 and Table A4). 

Possible associations between MED-DQI dietary quality index scores and gut microbiota were analysed in PWH (Figure 3 and Table A5), showing a negative correlation between MED-DQI with *Lachnospira* (Rho = −0.42; *p*-value *=* 0.028) and a positive correlation with *Bacteroides* (Rho = 0.48; *p*-value = 0.012). Lower MED-DQI scores indicate better quality diet and are associated with higher abundance of *Lachnospira*, and lower levels of *Bacteroides*.

We explored the possible correlation of *Lachnospira* and *Bacteroides* with nutrients and foods in (Figure 4 and Figure 5). We observed a positive correlation between *Lachnospira* and cellulose (Rho = 0.49 and *p*-value = 0.008), fructose (Rho = 0.49 and *p*-value = 0.009), dietary fibre (Rho = 0.48 and *p*-value = 0.012), and glucose (Rho = 0.41 and *p*-value = 0.034). We observed a negative correlation between *Bacteroides* and glucose (Rho = −0.51 and *p*-value = 0.006), malic acid (Rho = −0,46 and *p*-value = 0.015), citric acid (Rho = −0.41 and *p*-value = 0.032), olive oil (Rho = −0.39 and *p*-value = 0.044) and vegetables (Rho = −0.38 and *p*-value = 0.048).

### 3.5. Association between Dietary Pattern and Inflammatory Biomarkers

To further investigate the potential role of the diet on systemic makers of clinical progression, we analysed the associations between the dietary pattern and inflammatory biomarkers. We found that participants with a MEL pattern, exhibited reduced D-dimer (*p*-value = 0.050) and sTNFR2 (*p*-value = 0.049) levels compared to those with a WEL pattern (Figure 6 and Table A6).

### 3.6. Association between Dietary Pattern and Gut Microbiota in MSM

Additionally, significant differences were observed in the levels of *Erysipelotrichaceae* (*p*-value = <0.001) in MSM participants with MEL pattern versus MSM participants with WEL pattern, being more abundant in the latter, accounting for an approximate two-fold increase (Figure 7 and Table A7).

We explored in the whole MSM the possible correlation of Erysipelotrichaceae with nutrients and foods in (Figure 8). We observed a negative correlation with vitamin E (Rho = −0.9 and *p*-value *=* 0.037), fish (Rho = −0.76 and *p*-value < 0.001), zeaxanthin (Rho = −0.55 and *p*-value = 0.019), and a positive correlation with meat (Rho = 0.75 and *p*-value < 0.001).

In MSM with MEL pattern (Figure 9), we observed a negative correlation with the carotenoide zeaxanthin (Rho = −0.61 and *p*-value = 0.027), fish (Rho = −0.58 and *p*-value *=* 0.038) and with eicosanpentanoic Acid (Rho = −0.55 and *p*-value = 0.049).

In contrast, in MSM with WEL pattern (Figure 10) we observed a negative correlation with vitamin E (Rho = −0.9 and *p*-value = 0.037), fruit (Rho = −0.9 and *p*-value *=* 0.037) and dietary Fibre (Rho = −0.9 and *p*-value *=* 0.037).

### 3.7. Association between Dietary Pattern and Inflammatory Biomarkers in MSM

Analysing the data by risk group, we observed that in MSM participants with MEL pattern, there were s differences in D-dimer (*p*-value = 0.026) with mean levels almost half of those found in participants with WEL pattern (Figure 11 and Table A8).

## 4. Discussion

In this study in PWH, we found evidence of an intersection between dietary habits, the microbiota and systemic inflammation. We found that compared to participants with a WEL pattern, those with MEL pattern exhibited increased abundance of *Lachnospira* and lower levels of D-dimers and sTNFR2, and among MSM also a lower Erysipelotrichaceae abundance and lower D-dimers. While it is now assumed that Western and Mediterranean dietary patterns and dietary fibre intake play an important role in modulating gut microbiota [20,21,22,23,24,25], the information in the setting of specific chronic diseases such as HIV remains scarce.

*Lachnospira* genus is a producer of short-chain fatty acids (SCFA), especially butyrate production, which generates energy for colonocytes, which will induce enterocyte binding proteins, promoting intestinal barrier function and favouring the increase of intestinal regulatory T cells that mitigate inflammation [26]. Our findings regarding the association between total fibre intake and *Lachnospira* abundance are in keep with previous reports outside HIV infection [27,28]. In our study, a higher abundance of *Lachnospira* and dietary fibre intake was associated with a MEL dietary pattern, and a positive correlation was found between *Lachnospira* and dietary fibre. In addition, a higher carbohydrate intake (%) was observed in the MEL versus WEL pattern, although no correlation of the latter with *Lachnospira* was observed.

There was also a negative correlation between the MED-DQI with *Lachnospira* and a positive correlation with *Bacteroides*. This latter result is consistent with numerous published studies in the general population indicating that *Bacteroides* dominated microbiota [20,29], that is associated with long-term dietary patterns, and associated with a higher abundance of animal proteins and fats characteristic of the Western pattern [30,31].

In an exploratory analysis in PWH of this study, we observed a positive correlation between *Lachnospira* and digestive carbohydrates like fructose and glucose and non-digestive carbohydrates like dietary fibre. We also observed a negative correlation between *Bacteroides* and glucose, organic acids like malic and citric acid, olive oil, and vegetables.

Furthermore, we found a lower Erysipelotrichaceae abundance in MSM with MEL pattern. Although until recently, the Erysipelotrichaceae family of bacteria was not considered to play a significant role in health and disease. However, an increasing number of studies have been published attributing a potential role in numerous diseases and inflammatory processes [32] and linking it to a high-fat Western diet [33]. Notably, one study found that the relative abundance of *Erysipelotrichix*, a bacterium belonging to the Erysipelotrichaceae family, correlated positively with levels sTNFR2 [32] in PWH on ART. Other studies in animal models showed a positive correlation between against the abundance of Erysipelotrichaceae levels and cholesterol metabolites [34,35] and that supplementation with antioxidants (quercetin), which are typically abundant in the MEL pattern, inhibited the growth of against the abundance of Erysipelotrichaceae [36].

In an exploratory analysis restricted to MSM in this study we observed a negative correlation between vitamin E, zeaxanthin and fish consumption with Erysipelotrichaceae abundance and a positive correlation with meat. For the case of MSM with MEL we observed there is a negative correlation with fish and eicosapentaonic acid (EPA) w-3 fatty acids found in oily fish with anti-inflammatory action and is clinically relevant beneficial [37,38] and with the carotenoide zeanxanthin. MSM with WEL pattern showed a strong negative correlation of Erysipelotrichaceae with antioxidants intake like vitamin E, dietary fibre and fruit consumption.

Importantly, we found differences in the inflammatory biomarkers D-dimmer and sTNFR2 among PWH according to the dietary pattern followed, with levels in MEL being much lower than in WEL. sTNFR2 is a proinflammatory cytokine that is used as a marker of atherogenesis and is increased in PWH despite effective ART [37], as well as in other pro-inflammatory diseases [39,40,41]. D-dimer is a marker of fibrin degradation and its levels are often increased in PWH and denote a pre-thrombotic state that may lead to clinical thrombosis [42]. Importantly, higher levels of inflammatory biomarkers during treated HIV have consistently been linked to an excess risk of comorbidities during HIV treatment and suggested as a contributing risk factor [43]. As chronic inflammation has become a stubbornly elusive target in HIV, pursuing the effect of dietary interventions deserve further investigations. Our findings linking the MEL pattern with microbiome shifts and lower inflammation provide further support to recent intervention studies showing that adherence to the Mediterranean diet or particular components such as extra virgin olive oil is a feasible intervention to mitigate chronic inflammation in PWH [44,45,46].

Our study also has several limitations that must be taken into considerations when interpreting our results. First, we only assessed a limited number of participants, so we might have lacked statistical power to detect differences between groups in some analyses. Second, the observational design prevented us to assess causal relationships. However, in the context of the current state of the art, we can safely assume that dietary changes should explain the differences in the microbiome composition and inflammatory biomarkers, instead of the opposite.

## 5. Conclusions

In this observational study in PWH, compared to WEL pattern, MEL pattern was characterised by a higher intake of vegetables, cereals and legumes and a high intake of fibre, as well as a lower intake of sugars, meat and potatoes. MEL pattern determined higher *Lachnospira* abundance and a better score on the MED-DQI food quality index, which correlated directly with *Lachnospira* abundance and indirectly with *Bacteroides* abundance. In addition, MSM with MEL pattern exhibited lower abundance of Erysipelotrichaceae than MSM with a WEL pattern, a family of bacteria implicated in numerous inflammatory processes and which is more prevalent in Western diets characterised by a high intake of saturated fat and cholesterol and a low intake of dietary fibre, vitamins and antioxidants. Finally, MEL pattern was associated with decreased levels of D-dimer and sTNFR2, suggesting beneficial effects on long-term clinical outcomes.

Our study supports the notion that the dietary interventions should be further pursued as a feasible target to decrease inflammation in treated HIV. Larger studies with larger sample sizes assessing dietary interventions are justified to further inform the best strategies to improve chronic inflammation in PWH.

## Figures and Tables

**Figure 1 nutrients-14-01221-f001:**
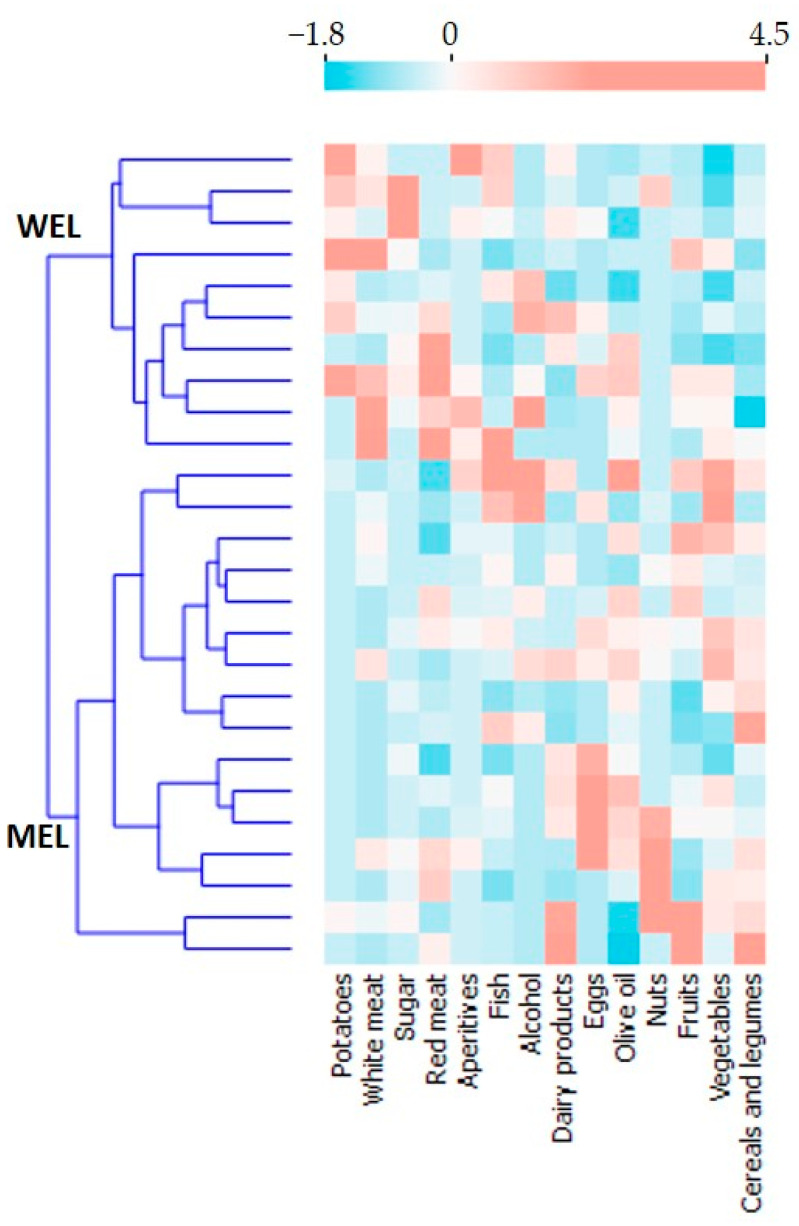
Heatmap z-score variables 14 food groups.

**Figure 2 nutrients-14-01221-f002:**
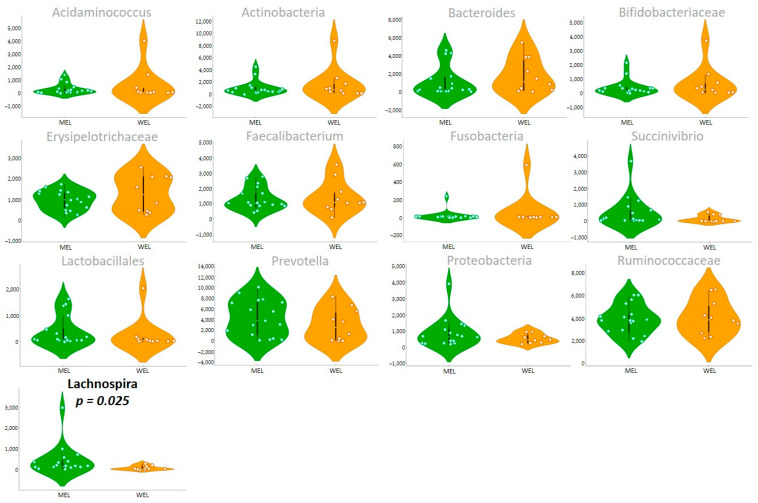
Gut microbiota by dietary pattern.

**Figure 3 nutrients-14-01221-f003:**
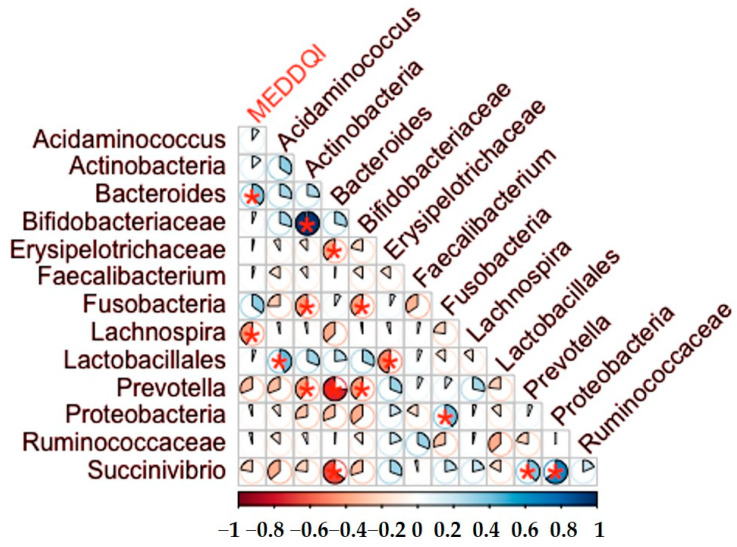
Heat map of correlations between MEDDQI and bacterial counts. The pie charts represent the magnitude of each individual Spearman Rho correlation coefficient in a color gradient from red (Rho −1) to blue (Rho +1). Correlations with a *p*-value < 0.05 are marked with a red asterisk.

**Figure 4 nutrients-14-01221-f004:**
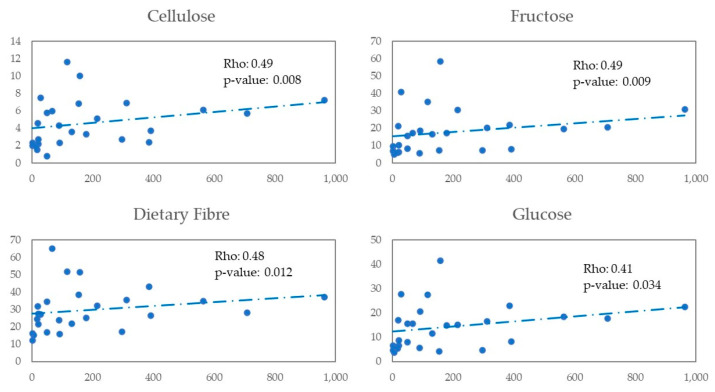
Scatter plot X—*Lachnospira*, Y—Nutrients/Food.

**Figure 5 nutrients-14-01221-f005:**
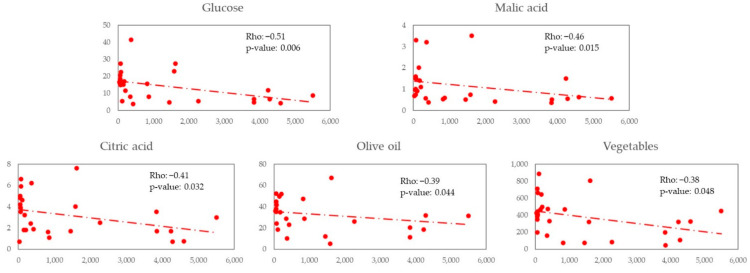
Scatter plot X—*Bacteroides*, Y—Nutrients/Food.

**Figure 6 nutrients-14-01221-f006:**
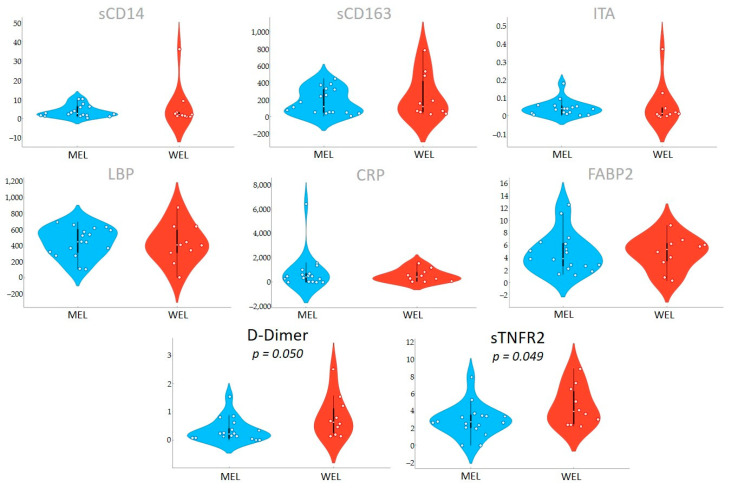
Inflammatory biomarkers by dietary pattern.

**Figure 7 nutrients-14-01221-f007:**
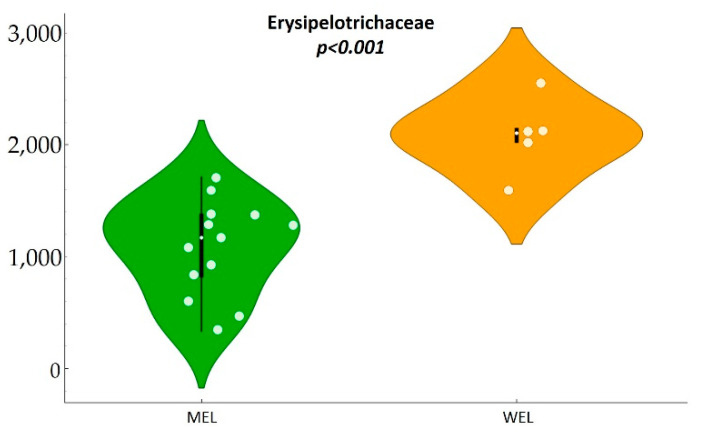
Erypelotrichaceae by dietary pattern in MSM.

**Figure 8 nutrients-14-01221-f008:**
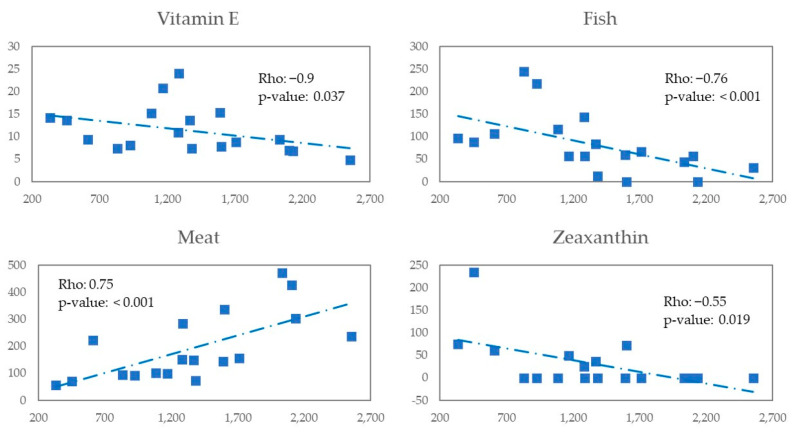
Scatter plot X—Erysipelotrichaceae, Y—Nutrients/Food for all MSM.

**Figure 9 nutrients-14-01221-f009:**
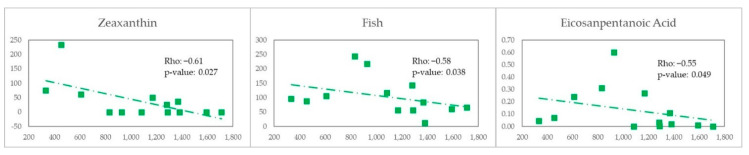
Scatter plot X—Erysipelotrichaceae, Y—Nutrients/Food for MSM-MEL.

**Figure 10 nutrients-14-01221-f010:**
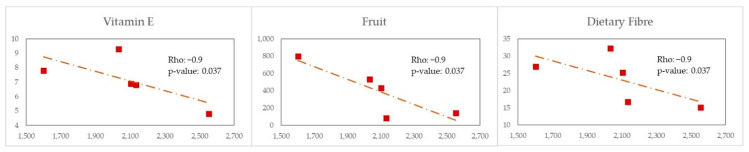
Scatter plot X—Erysipelotrichaceae, Y—Nutrients/Food for MSM-WEL.

**Figure 11 nutrients-14-01221-f011:**
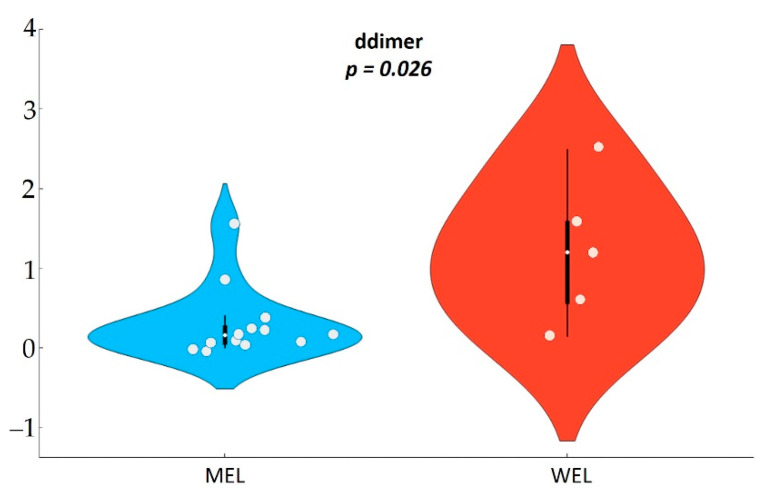
D-dimer biomarker in MSM by dietary pattern.

**Table 1 nutrients-14-01221-t001:** Sample characteristics by dietary pattern.

	MEL (*n* = 17)	WEL (*n* = 10)	Total (*n* = 27)	*p*-Value
Age (Years)	46.29 ± 12.16	52.20 ± 6.75	48.48 ± 10.74	0.172
Sex, *n* (%) ^(X2)^				0.303
Female	1 (5.9)	2 (20.0)	3 (11.1)	
Male	16 (94.1)	8 (80.0)	24 (88.9)	
Risk factor, *n* (%) ^(X2)^				0.329
MSM	13 (76.5)	5 (50.0)	18 (66.7)	
HTX	1 (5.9)	2 (20.0)	3 (11.1)	
IDU	3 (17.6)	3 (30.0)	6 (22.2)	
Ethnicity, *n* (%) ^(X2)^				0.818
Caucasian	13 (76.5)	8 (80.0)	21 (77.8)	
Latin American	3 (17.6)	1 (10.0)	4 (14.8)	
Sub-Saharan African	1 (5.9)	1 (10.0)	2 (7.4)	
Anthropometric dates				
Weight (kg) ^(U)^	75.90 ± 11.29	74.86 ± 12.90	75.51 ± 11.67	0.821
Size (cm)	174.71 ± 6.91	170.70 ± 4.76	173.22 ± 6.41	0.119
BMI (kg/m^2^) ^(U)^	24.85 ± 3.25	25.75 ± 4.79	25.19 ± 3.82	0.633
Use of antibiotic in past 6 months, *n* (%) ^(X2)^	3 (17.6)	3 (30.0)	6 (22.2)	0.387
On triple ART, *n* (%)	17 (100.0)	10 (100.0)	27 (100.0)	0.530
INSTI-based	10 (59)	5 (50)	15 (56.0)	
NRTI-based	6 (35)	3 (30)	9 (33.0)	
PI-based	1 (6)	2 (20)	3 (11.0)	
Undetectable HIV RNA	17 (100.0)	10 (100.0)	27 (100.0)	-
Use of drugs with anti-inflammatory effects				
Atorvastatine	1 (6.0)	0 (0)	1 (6.0)	-

Results are presented as mean ± SD when variables are continuous, or *n* (%) when categorical. Significant differences between dietary patterns (*p*-value < 0.05) are indicated in bold. ^(U)^: *p*-value calculated with the Mann-Witney U test. ^(X2)^: *p*-value calculated with Chi-square test. MSM: Men who have sex with men; HTX: Heterosexuals; IDU: Ex-injection drug users; BMI: Body Mass Index; ART: antiretroviral therapy.

**Table 2 nutrients-14-01221-t002:** Intake (g/day) of each food group per dietary pattern.

	MEL (*n* = 17)	WEL (*n* = 10)	Total (*n* = 27)	*p*-Value
**Cereals and legumes**	**279.87 ± 100.31**	**174.28 ± 49.27**	**240.76 ± 98.65**	**0.001**
**Vegetables**	**455.34 ± 223.45**	**257.02 ± 179.31**	**381.89 ± 226.67**	**0.042**
Fruits ^(t)^	463.85 ± 363.98	307.13 ± 218.93	405.80 ± 322.59	0.175
Dairy products	311.18 ± 195.16	252.87 ± 133.03	289.58 ± 174.32	0.547
Nuts	11.75 ± 17.97	2.40 ± 7.59	8.29 ± 15.48	0.059
**Meat ^(t)^**	**120.29 ± 75.29**	**288.80 ± 156.22**	**182.70 ± 137.16**	**0.008**
**White meat**	**23.92 ± 35.64**	**117.74 ± 103.48**	**58.67 ± 81.36**	**0.004**
Red meat	96.37 ± 73.29	171.06 ± 104.48	124.03 ± 91.84	0.056
Fish	106.45 ± 90.45	122.02 ± 125.45	112.22 ± 102.67	0.900
Eggs	37.66 ± 48.86	14.64 ± 22.36	29.13 ± 42.08	0.189
Sugar	6.51 ± 8.30	44.14 ± 67.02	20.45 ± 44.05	0.071
Olive oil ^(t)^	34.84 ± 15.92	28.04 ± 13.09	32.32 ± 15.05	0.265
Alcohol	7.58 ± 12.29	10.97 ± 15.49	8.83 ± 13.37	0.620
Aperitives	5.91 ± 11.38	28.41 ± 49.95	14.24 ± 32.65	0.159
**Potatoes**	**2.55 ± 8.29**	**75.00 ± 71.40**	**29.38 ± 55.48**	**0.001**

Results are presented as mean ± SD. Significant differences between dietary patterns (*p*-value < 0.05) are indicated in bold. ^(t)^: *p*-value calculated with *t*-test for independent samples. Other variables *p*-value calculated with Mann–Whitney U test.

## Data Availability

The data presented in this study are available upon reasonable request to the corresponding author. Some restrictions might apply due to ethics limitations.

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
