# Peer review of "Relationship of Diet to Gut Microbiota and Inflammatory Biomarkers in People with HIV"

_nutrients, 2022, doi:10.3390/nu14061221_

Round 1

Reviewer 1 Report

In this interesting manuscript, Manzano et al. showed that a Mediterranean-like (MEL) diet is associated with more beneficial microbes in the gut microbiota of Spanish PWH, compared to a western diet. They found more Lachnospira bacteria in those with a MEL diet. Also, PWH with a MEL diet had lower inflammation in the blood with lower D-dimer and TNFalpha, compared to those with a WEL diet. When looking at MSM, they found lower levels of Erysipelotrichaceae in those with a MEL diet, and also found lower D-dimer in MSM with a MEL diet compared to MSM with a WEL diet.

The manuscript is interesting and well written. Its message is clear and the data convincing.

Here are my suggestions to improve it.

  • I would present section 3.5 and figure 8 before section 3.4.1, to group all findings in MSM together and ease reading.
  • Violin plots are not recommended for a small number of data point. Can the author provide figures with plots showing individual datapoint (fig 2, 4 8 and 9)?
  • Can the author describe and compare the groups for use of non-ARV medications that could influence inflammation and gut microbiota composition?
  • Can the authors comment on the absence of difference of Lachnospira abundance between MEL and WEL in MSM? what is their hypothesis?
  • In Fig 5 and 6, the authors explore the link between abundance of Erysipletrichaceae and certain nutrients in MSM. A similar analysis would be interesting for Lachnospira and possibly Bacteroides as those were linked with MED-DQI, in all participants.
  • Lachnospira are known producers of SCFA, as indicated by the authors. The addition of SCFA measurements in plasma samples would be a great addition to corroborate gut microbiota differences observed in this study.
  • In the discussion, the author state that “Lachnospira abundance was associated with a MEL dietary pattern and the differences compared to a WEL pattern were driven by a higher intake of carbohydrates (%) and dietary fibre in the MEL pattern.” Can the author clarify the influence of each nutrient on the difference of Lachnospira abundance between the 2 groups?
  • Addition of references in the introduction would strengthen the indicated link between diet, gut microbiota, metabolism and health (PMID 28388917 and 33432175 for instance).

Minor comments:

  • The manuscript should be checked for typo and grammatical mistakes. For instance: antriretroviral line 72, sum of percentages not reaching 100% line 197, ref missing line 221, “also” line 259, “indirectly” line 322.
  • When appropriate, please avoid using “patient” and choose “participant”.

Reviewer 2 Report

In this study the authors aimed to analyse the dietary patterns and dietary quality in a group of 27 HIV-infected patients and to correlate the nutritional parameters with the gut microbiota composition and inflammatory biomarkers. The study seems interesting and the results are nicely presented. However, several issueas need to be clarified.

Major issues:

  1. Please give more details on the baseline characteristics of the study group (age, sex were there only male patients?, cART scheme)
  2. When were the stool samples colected? Simoultaneously with the blood samples? At the same time the dietary interview was collected?
  3. Were there any associations between the gut microbiota and patien'ts BMI ora cART used? Please discuss.

Minor issues:

  1. Abstract, Conclusions: "butyrate producer that is associated with reduced intestinal inflammation,  and protects against Erysipelotrichaceae abundance" - this statement is not a conclusion from this study, it should be rather removed to the Introduction section.
  2. Please explain the abbreviations at the first time they are used in the main text (line 61) and please explain the abbreviations used in the table (Table 1) in a caption

Round 2

Reviewer 2 Report

The authors have answered all the queries. In my opinion, the manuscript conveys now a clearer message and is now suitable for publication. Thank you.